# Comfortable and Convenient Turning Skill Assessment for Alpine Skiers Using IMU and Plantar Pressure Distribution Sensors

**DOI:** 10.3390/s21030834

**Published:** 2021-01-27

**Authors:** Seiji Matsumura, Ken Ohta, Shin-ichiroh Yamamoto, Yasuharu Koike, Toshitaka Kimura

**Affiliations:** 1Sports Brain Science Project, NTT Communication Science Laboratories, Nippon Telegraph and Telephone Corporation, Atsugi 243-0198, Japan; ohta@sports-sensing.com (K.O.); toshitaka.kimura.kd@hco.ntt.co.jp (T.K.); 2Department of Information Processing, Tokyo Institute of Technology, Yokohama 226-8503, Japan; 3Department of Bio-science and Engineering, Shibaura Institute of Technology, Saitama 337-8570, Japan; yamashin@se.shibaura-it.ac.jp; 4Institute of Innovative Research, Tokyo Institute of Technology, Yokohama 226-8503, Japan; koike@pi.titech.ac.jp

**Keywords:** sports performance, skill assessment, inertial measurement units (IMU), plantar pressure distribution sensors, feature detection, ski, actual field evaluation

## Abstract

Improving ski-turn skills is of interest to both competitive and recreational skiers, but it is not easy to improve on one’s own. Although studies have reported various methods of ski-turn skill evaluation, a simple method that can be used by oneself has not yet been established. In this study, we have proposed a comfortable method to assess ski-turn skills; this method enables skiers to easily understand the relationship between body control and ski motion. One expert skier and four intermediate skiers participated in this study. Small inertial measurement units (IMUs) and mobile plantar pressure distribution sensors were used to capture data while skiing, and three ski-turn features—ski motion, waist rotation, and how load is applied to the skis—as well as their symmetry, were assessed. The results showed that the motions of skiing and the waist in the expert skier were significantly larger than those in intermediate skiers. Additionally, we found that the expert skier only slightly used the heel to apply a load to the skis (heel load ratio: approximately 60%) and made more symmetrical turns than the intermediate skiers did. This study will provide a method for recreational skiers, in particular, to conveniently and quantitatively evaluate their ski-turn skills by themselves.

## 1. Introduction

Alpine skiing, in which skiers make turns to descend a snow-covered slope, is one of the most popular winter sports. Improving one’s skill in making ski turns comfortably is beneficial for both competitive and recreational skiers. Turns performed by a highly skilled skier are, on a flat surface, for example, symmetric [1,2,3]. Such a skilled ski turn requires accurate and rhythmic control of the skis—proper load distribution and active changes in posture are important. For recreational skiers, the main method to improve skiing skills is to receive instruction at a ski school because it is difficult to learn these skills effectively or to grasp skill level objectively on one’s own. A learning method that shows the relationships between changes in load, posture, and skiing motion would be convenient to facilitate learning skiing effectively on one’s own; therefore, a minimalistic assessment methodology that can extract these relationships for skill improvement is required.

To capture data to evaluate ski-turn skills in terms of the relationship between the physical movement of a skier and the interaction of the skis with the snow’s surface, photos, videos, and inertial measurement units (IMUs) are commonly used [4,5,6,7,8,9,10]. In a previous study, multiple IMUs were attached to the body of a professional alpine skier to measure body movements during skiing [11]. In another study, the kinematics of a skier’s center of mass during alpine ski racing were estimated using IMUs [12]. Plantar pressure distribution sensors are commonly used to measure load distribution on skis [13,14,15], for example, the distribution of plantar pressure has been measured during a turn in an alpine skier [16] and the relationship between plantar pressure distribution and slope steepness in alpine skiing has been investigated [17]. Furthermore, there have also been several previous ski-turn skill assessment studies. Motion analysis of ankle joints during ski turns has been performed by simultaneously using IMUs and plantar pressure distribution sensors [18]; skiing styles, such as skidding or carving turns, of alpine skiing have been evaluated using IMUs and the global navigation satellite system [19], and physical skills have been assessed from lower extremity muscle activity during skiing [20]. Although previous studies have provided detailed analyses and evaluations, these methods frequently use large and expensive technology, thus limiting their use by recreational skiers.

However, inexpensive and convenient (such as mobile or wearable) devices have been commercialized for recreational skiers. One such product assesses skiing skills by using IMUs that are attached to the skis and chest [21]. There is also a product that uses an inertial sensor and a load sensor attached to skis to evaluate skiing skills [22]. Although these products make use of a small number of sensors, useful gadgets, and artificial intelligence to assess skiing skills, the relationship between body movement and the motion of the skis is still unclear. This is because these products provide a general evaluation without this relationship. Therefore, it is unclear, for ordinary skiers, what they should do to improve their skiing skills.

The main purpose of this study is to propose a convenient methodology that is able to assess ski-turn skills using the features of the skill, especially for non-professional athletes. To extract the features, we measured and compared the ski-turn movements of advanced and intermediate skiers. For convenience, small wearable IMUs were used. Mobile plantar pressure distribution sensors were also used to capture the relationship between the load on the skis and the ski motion. This experiment was conducted on an actual piste, and the motion and load during ski turns were measured.

## 2. Materials and Methods

### 2.1. Participants

One expert skier (male) and four intermediate skiers (3 males, 1 female) participated in this study. The expert skier was officially ranked at Technical Prize by the Ski Association of Japan; see Figure 1. He represented the ideal skilled skier in this study. Three of the intermediate skiers (2 males, 1 female) were ranked at 1st degree by the Ski Association of Japan. One intermediate skier did not have any technical certifications; however, since he skied at least 30 days per year for the previous 10 years, he was categorized as an intermediate skier. The participants were given an information sheet that outlined the general purpose of the study and were informed that they could withdraw from the study at any time without penalty. All methods employed in this study were approved by the Ethics and Safety Committees of NTT Communication Science Laboratories and were in accordance with the Declaration of Helsinki (protocol code: H30–002; date of approval: 28 March 2018).

### 2.2. Measurement Sensors

Sensors—IMU (Sports Sensing Co. Ltd., Fukuoka, Japan) and plantar pressure distribution (pedar*^®^* system, Novel GmbH, München, Germany)—were used with a temporal synchronization device; see Figure 2. Each sensor was monitored and controlled wirelessly, with specific software, via laptop. In each trial, first, the laptop computer sent a wireless signal to the sensors to place them in standby mode. Then, the time synchronizer sent a wireless signal to all sensors to simultaneously trigger the start of recording. The synchronization error was less than 0.1 ms; this error was sufficiently small for human motion measurement. During measurements, each sensor independently recorded data for 2 min and then automatically stopped. When a skier was within the Bluetooth*^®^* wireless communication range, the sensor measurement status could be monitored on the laptop computer. Data generated by each sensor were time-stamped and stored in the sensor’s own memory. After the experiment, the data were extracted from each sensor’s memory for offline analysis.

The IMUs had a three-dimensional accelerometer and a three-dimensional gyroscope and were waterproof. Quaternions were calculated automatically from the IMUs and were used to estimate the orientation of the plane on which the sensor was placed. The sampling frequency of each IMU sensor was set to 1000 Hz. Three IMUs were used: two were placed on the skis before binding and one on the skier’s hip at the L5–S1 position of the vertebral column. The sensors on the skis captured the movement of the skis and the sensor on the waist captured the movement of the body. Because these sensors were small, they did not interfere with the skier’s movement as they skied.

The plantar pressure distribution sensors had a similar shape to shoe insoles and were installed inside the ski boots. This system also had a mobile control unit. The sensors’ data were transmitted by cable, running along the foot, to the control unit attached to the waist and were stored in the memory; therefore, the participants could ski without any restriction of movement. The sensors divided the sole of the foot into 99 regions and recorded the pressure in each region. The sampling frequency of the plantar pressure distribution sensors was set to 100 Hz.

### 2.3. Procedure

The experiment was conducted on an actual piste. The piste was flat, the average slope was 25°, and the conditions were not icy but the snow had slightly melted (similar to snow in early spring). Each participant was asked to ski for two runs and, on each run, to ski down the piste in six parallel turns while their data were recorded. In this study, to eliminate the effects of different skis on turn performance, participants used the same skis (Salomon, Rush 165 cm, 2019 model), which were provided to each participant for use during the experiment.

Before the start of the measurements, participants waited on a flat area at the top of the slope. Once the measurement began, the participants held still for 5 s to calibrate the sensors, standing in a neutral position—relaxed, with knees slightly bent; see Figure 3. This allowed changes from the data recorded in the neutral position to be examined during analysis. After calibration, the participants proceeded to the slope and completed their run, after which they returned to the first waiting area at the top of the slope using the chair lift until it was time for their next run. All sensors automatically stopped recording 2 min after the start of measurement.

### 2.4. Data Analysis

Based on our previous work [23], in this study, three ski-turn features were analyzed offline: ski motion, i.e., edge rotation angular velocity, was measured using an IMU on each ski; waist motion, i.e., waist rotation, was measured using an IMU on the waist; and plantar force, i.e., load applied to the skis, was measured using plantar pressure distribution sensors. Furthermore, to assess general ski-turn skill, the left–right symmetry of the turns was analyzed. For simplicity, we focused on the outside leg during ski turns, i.e., the right leg on left turns and the left leg on right turns.

In preprocessing, changes in angular velocity about the *y*-axes of the IMUs attached to the skis were used to partition the data; see Figure 4. The *y*-axes of the IMUs pointed in the direction of travel (i.e., along the length of the ski), and thus, the angular velocity about these axes represented the edge rotation speed in a turn; see Figure 2. The orientation of the other axes of the IMUs and how they correspond with the orientation of the ski or the body is also shown in Figure 2. Using peak detection with the *y*-axis angular velocity data, wherein the peaks indicate the instant of the change in motion direction during the ski turn, the data between consecutive peaks, which included the end of the previous turn and the beginning of the next turn, were extracted. Therefore, the interval between those peaks corresponds to one turn. In the coordinate system used in this experiment, angular velocity time-series data from positive to negative peaks corresponded to a left turn, and those from negative to positive peaks corresponded to a right turn. We eliminated the first and last turns in each trial as they included starting and stopping motions in addition to turning motions. Thus, four sets of each turn-direction data were analyzed.

Edge rotation angular velocity about the *y*-axis of the IMUs attached to skis was used to measure ski motion. The angular velocity time-series were preprocessed with a 5-Hz low-pass filter, and each turn was extracted as one interval. Third-order spline interpolation was then applied to the filtered data and resampled to 3000 points to yield the same number of data points for each turn interval, which allowed us to treat intervals of the same turn direction as the same type of interval. The resampled data during one interval were first evaluated qualitatively. Then, the sum of the squares of angular velocity *v* during each interval was calculated and evaluated as the quantitative feature for edge rotation:(1)$$Feature_{skis}=\underset{n}{\sum }v_{n}^{2}$$
where *n* is the number of data points.

For waist rotation, we used data from the IMU worn at the waist to measure changes in posture. Quaternions outputted from the sensor were used, and the rotation was represented as the change from the position of the unit normal vector (with the sensor center as the origin) during calibration to that of the unit vector at each time point. Therefore, the change in the posture was expressed as coordinates on a unit sphere at a distance of 1 norm from the origin. Third-order spline interpolation was applied, and the data for each turn (based on the partitions determined from the synchronized ski IMU data) were resampled to 3000 points to yield the same number of data points for each turn interval. Resampled data during one interval were first evaluated qualitatively. Then, the peak-to-peak value of posture change during one interval was calculated and evaluated as a feature quantifying waist rotation,
(2)$$Feature_{waist}=x_{max}-x_{min}$$
where *x_max_* is the maximum coordinate value during a turn and *x_min_* is the minimum coordinate value.

Plantar pressure distribution sensors were used to record plantar forces in regions of the foot. Sensor placement was divided into two regions—toe and heel; see Figure 5. The force perpendicular to the sensors’ surfaces was determined in each region; see Equation (3). This was defined as the load applied by the foot to the skis,
(3)$$F=\underset{i\;\in \;targeted\;sensor{}^{\prime }s\;IDs}{\sum }A_{i}P_{i}$$
where *F* is the force generated from the target area (the load to the skis), *A* is the area of each sensor that divided the foot into 99 segments (the known sensor-specific value), and *P* is the pressure measured by each sensor. Then, load data extracted at each turn were preprocessed with a 5-Hz low-pass filter. Third-order spline interpolation was applied to the filtered data, and the data were resampled to 500 points to yield the same number of data points for each turn. To examine whether the load was applied by the toe or the heel to the skis, the heel load ratio was calculated as the heel load divided by the entire load. Heel load ratio time-series data during one interval were evaluated qualitatively, and then, the mean heel load ratio during one interval was calculated and evaluated as a quantitative feature of the load applied to the skis:(4)$$Feature_{plantar\;force}=\left(\frac{\bar{F_{heel}}}{\bar{F_{toe}}+\bar{F_{heel}}}\right)\times 100$$
where $\bar{F_{toe}}$ is the mean force generated from the toe region during a turn, and $\bar{F_{heel}}$ is the mean force generated from the heel region.

For the three quantitative features above, the differences between skiers and turn directions were analyzed using a two-way repeated measures mixed model (between-subjects factor: skiers [5]; within-subject factor: turn direction [2]) analysis of variance (ANOVA) followed by post-hoc Holm testing. JASP (version 0.14) statistical software was used for these analyses. Statistical significance was set at *p* < 0.05. Finally, the left–right symmetry was calculated for general ski-turn skill assessment. For the three quantitative features respectively, the mean value of the left-turn features divided by that of the right-turn features was calculated as the feature of the left–right symmetry,
(5)$$Turn\;symmetry=\frac{\mu_{left}}{\mu_{right}}$$
where $\mu_{left}$ is the mean value of a left-turn feature, and $\mu_{right}$ is the mean value of a right-turn feature.

## 3. Results

### 3.1. Ski Motion

Angular velocity time-series data showing edge rotation are shown in Figure 6a. The angular velocity of the expert skier had large extreme values, a large-magnitude slope at the beginning of the turn, was approximately constant in the middle, and had an opposite-direction large-magnitude slope at the end of the turn, whereas the angular velocities of some of the intermediate skiers had small-magnitude slopes at the beginning and end of the turns, and the angular velocity in the middle of the turn was not constant.

The quantitative feature of ski motion, the sum of the square of edge rotation angular velocity, is shown in Figure 6b. There was a significant simple main effect between skiers (F(4, 15) = 15.666, *p* < 0.001, partial η^2^ = 0.807), and the post-hoc tests showed that the sum of the square of edge rotation angular velocity of the expert was significantly greater than those of the intermediate skiers; see Table 1. The simple main effect between turn directions was not significant (F(1, 15) = 0.413, *p* = 0.530, partial η^2^ = 0.027), and there was no significant interaction between skiers and turn directions (F(4, 15) = 2.002, *p* = 0.146, partial η^2^ = 0.348). These results indicate that differences between the expert and intermediate skiers’ ski motion characteristics—i.e., changes in angular velocity at the beginning and the end of a turn, and constant during the middle of a turn—were significant.

### 3.2. Waist Rotation

Waist rotation time-series data are shown in Figure 7a. The expert skier gradually rotated from one direction to the other (maximum to minimum or vice versa), whereas the intermediate skiers had smaller waist rotation ranges. The quantitative feature of waist motion, the peak-to-peak difference in waist rotation, is shown in Figure 7b. There was a significant simple main effect between skiers (F(4, 15) = 14.267, *p* < 0.001, partial η^2^ = 0.792), and the post-hoc tests demonstrated that the peak-to-peak waist rotation of the expert was significantly greater than those of intermediate skiers 1, 3, and 4; see Table 2. The simple main effect between turn directions was not significant (F(1, 15) = 3.374, *p* = 0.086, partial η^2^ = 0.184), and there was also no significant interaction between skiers and turn directions (F(4, 15) = 0.815, *p* = 0.535, partial η^2^ = 0.179). 

### 3.3. Plantar Force

Heel load ratio time-series data are shown in Figure 8a. The expert skier demonstrated a tendency to load the skis from the toes to the heel and only slightly used the heel (heel load ratio: Approximately 60%), whereas this feature varied among intermediate skiers. Some intermediate skiers demonstrated a tendency similar to that of the expert skier, but others did not, loading much more with either the toe or the heel.

The mean heel load ratio during one interval is shown in Figure 8b. The expert skier’s heel load ratio was approximately 60% in both turns. There was a significant simple main effect between skiers (F(4, 15) = 42.708, *p* < 0.001, partial η^2^ = 0.919), and the post-hoc tests showed that the heel load ratio of the expert was significantly smaller than that of intermediate skier 1 and significantly greater than that of intermediate skier 4; see Table 3. The simple main effect between turn directions was not significant (F(1, 15) = 1.971, *p* = 0.181, partial η^2^ = 0.116), and there was also no significant interaction between skiers and turn directions (F(4, 15) = 0.808, *p* = 0.539, partial η^2^ = 0.177). These results indicate that the expert skier only slightly tends to use the heel to apply loads to skis.

### 3.4. Turn Symmetry as a General Assessment

The left–right symmetry results for ski-turn skill are shown in Figure 9. The expert skier demonstrated highly symmetric ski motion, waist rotation, and loading to skis with respect to left and right turns—the values of the symmetry features were approximately 1, whereas not all of the intermediate skiers demonstrated symmetric features.

## 4. Discussion

We used a comfortable method to capture ski-turn feature data on actual alpine pistes. Three features that could be intuitively interpreted by skiers were extracted to assess differences in skill levels. The results showed that the motions of ski and waist in the expert skier were significantly larger than those in intermediate skiers. Additionally, the expert skier showed approximately a 60% heel load ratio and a symmetric left–right ratio in the evaluated features. These results indicate that an expert skier largely moves their waist and only slightly tends to use the heel to apply loads to skis, resulting in an aggressive ski motion and highly symmetric left–right turns.

Because it has previously been reported that skilled skiers make symmetric turns [1] and, with our method, we found that the expert skier’s features demonstrated symmetry for ski motion, waist rotation, and load applied to the skis, it demonstrates that our method is able to effectively capture the features of an expert skier. Skilled skiers also have pronounced motions during turns [2]; our method captured this characteristic, demonstrated by the large range of the ski and waist feature values, and our heel load ratio results confirm the findings of a previous study that reported that skilled skiers load their skis using the heel rather than using the toe [24]. We focused on a small number of indices to simplify verification of the proposed method; therefore, further studies using other motions such as sliding of the skis or backward and forward motion of the waist, are required to comprehensively verify the effectiveness of our method.

We suggest that the proposed method can be useful for recreational skiers. Our method consists of three small IMUs and a pair of plantar pressure distribution sensors, which is similar to off-the-shelf configurations that can be used by recreational skiers [21,22]. In these off-the-shelf products, general evaluations are achieved with artificial intelligence and useful gadgets, but the relationship between ski and body motions is not sufficiently clear. Our method has captured the corresponding features of ski and body motions to evaluate ski-turn skills—skills that have been evaluated by expensive and complicated methods in previous studies [11,12,13].

The present study had a limitation that should be considered. Our proposed method mainly focused on sensors that could be worn comfortably, while data collection and analysis were limited to manual methods. However, from a practical perspective, development of an automatic evaluation system will be considered.

There are few previous studies that used a comfortable measurement method to extract ski-turn features and evaluate skier skill level. Our method can quantitatively assess ski-turn features in relation to those of an expert skier, allowing skiers to be aware of the differences between themselves and expert skiers. We believe that our technique could lead to the development of an effective and convenient training system, especially for self-taught recreational skiers.

## 5. Conclusions

In this study, we proposed a comfortable measurement method that can be easily and conveniently used by skiers themselves. This proposed method extracted features of ski-turn motion representative of differences in skill level. The measurement system consisted of three small IMUs and mobile plantar pressure distribution sensors that do not interfere with skiing. One expert skier and four intermediate skiers participated in this experiment, and ski motion, waist rotation, and how load is applied to the skis were measured on an actual piste. Three features that would be easy for skiers to interpret and that represent differences in skills were extracted and used to measure general ski-turn skill. The results indicated that an expert skier performs symmetric turns with aggressive motions. This study will contribute to the development of a method for recreational skiers to conveniently and quantitatively evaluate their own skiing skill level on actual pistes.

## Figures and Tables

**Figure 1 sensors-21-00834-f001:**
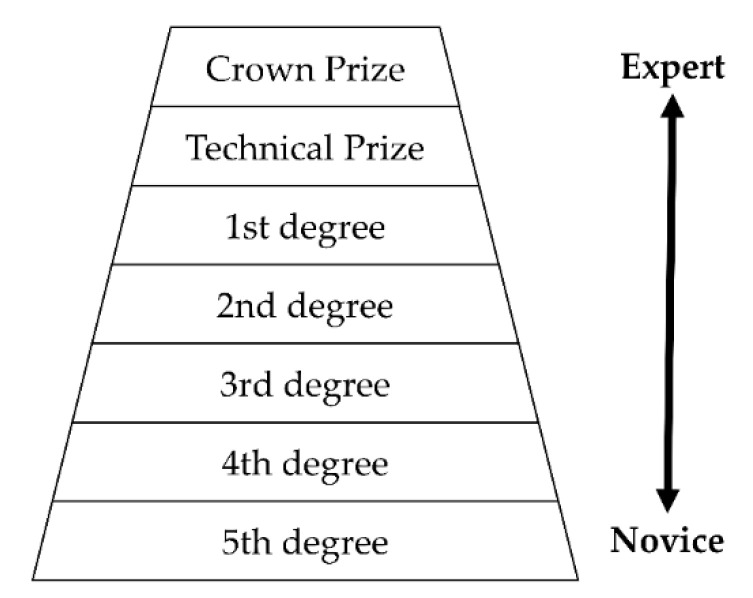
Ski Association of Japan technical certification ranks. Assessment is carried out by several judges who score the ability to ski in response to the situation and conditions, organization of the turn movements (positioning and edging), the ability to adapt to piste conditions (speed and rotation arc adjustment), and movement (balance, rhythm, and timing).

**Figure 2 sensors-21-00834-f002:**
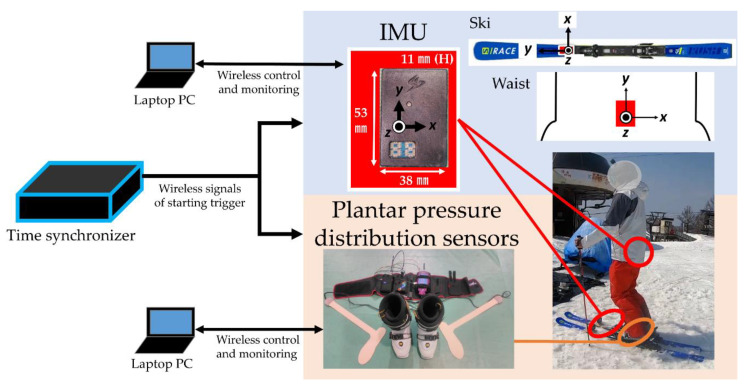
Inertial measurement units (IMUs), attached to both skis and the skier’s waist, and plantar pressure distribution sensors, in the ski boots, were synchronized; the sensors simultaneously started recording upon receipt of the start trigger signal from the time synchronizer. Each sensor was monitored and controlled wirelessly with specific software via the laptop.

**Figure 3 sensors-21-00834-f003:**
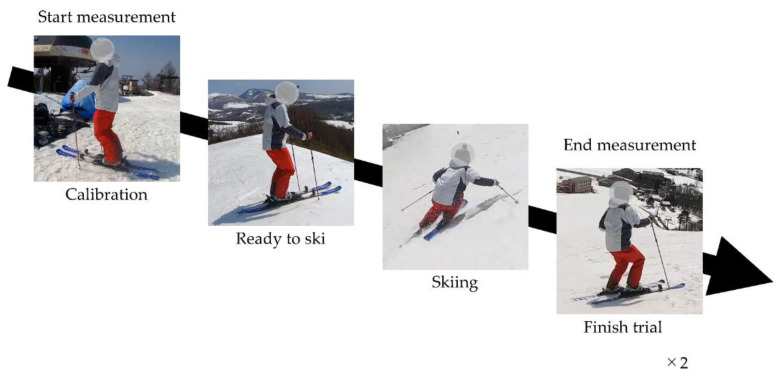
Experiment procedures. After the start of measurement, participants stood still for 5 s (for calibration) and then skied on the piste.

**Figure 4 sensors-21-00834-f004:**
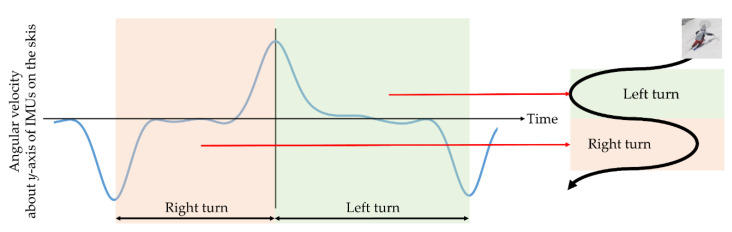
The change in angular velocity about the *y*-axis, indicating edge speed, of an inertial sensor installed on the skis was used to define one turn. In this experiment, right turns were defined as the interval from negative to positive peaks, and left turns were defined as the interval from positive to negative peaks.

**Figure 5 sensors-21-00834-f005:**
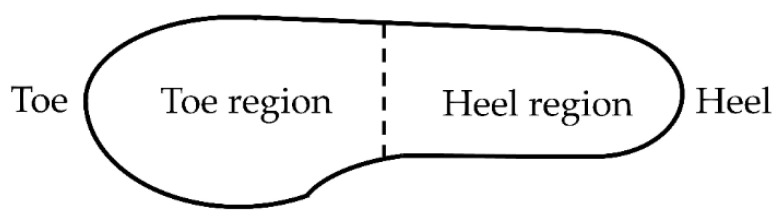
To investigate the tendency to load the skis, the foot was divided into two regions: toe and heel.

**Figure 6 sensors-21-00834-f006:**
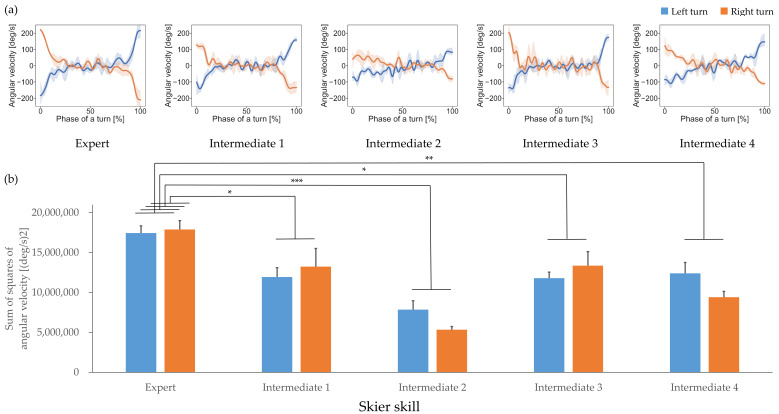
(**a**) Ski motion, demonstrated by edge rotation angular velocity, for each participant and (**b**) its representative quantitative feature. Blue and orange represent left and right turns, respectively. Statistical significance is indicated with * *p* < 0.05, ** *p* < 0.01, *** *p* < 0.001.

**Figure 7 sensors-21-00834-f007:**
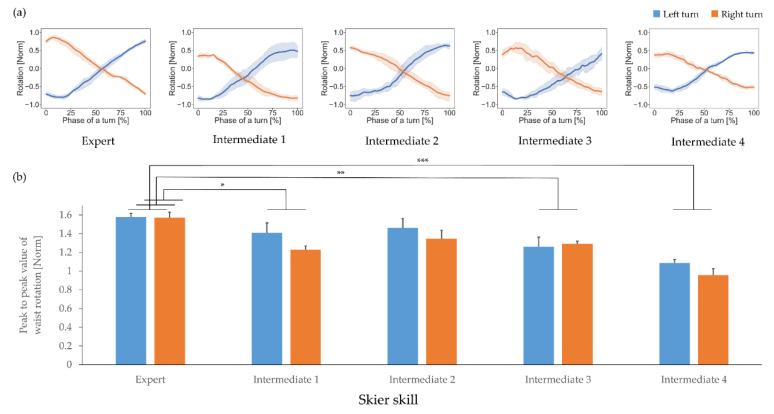
(**a**) Waist rotation for each participant, and (**b**) the quantitative feature representing waist rotation. Blue and orange represent the left and right turns, respectively. Statistical significance is indicated with * *p* < 0.05, ** *p* < 0.01, *** *p* < 0.001.

**Figure 8 sensors-21-00834-f008:**
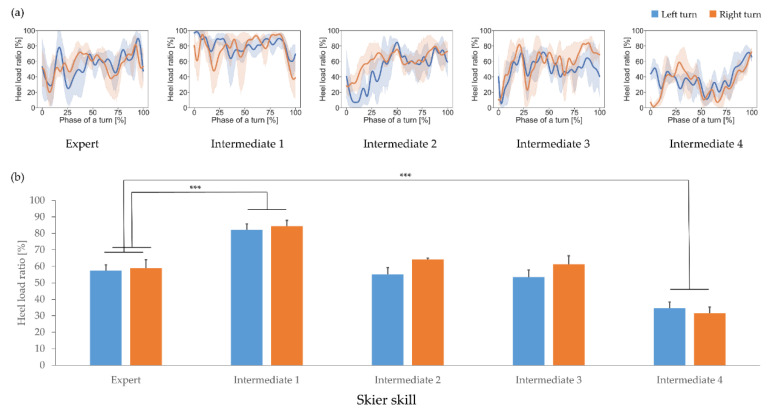
(**a**) Heel load ratio time-series data for each participant and (**b**) the mean heel load ratio. Blue and orange represent the left and right turns, respectively. Statistical significance is indicated with *** *p* < 0.001.

**Figure 9 sensors-21-00834-f009:**
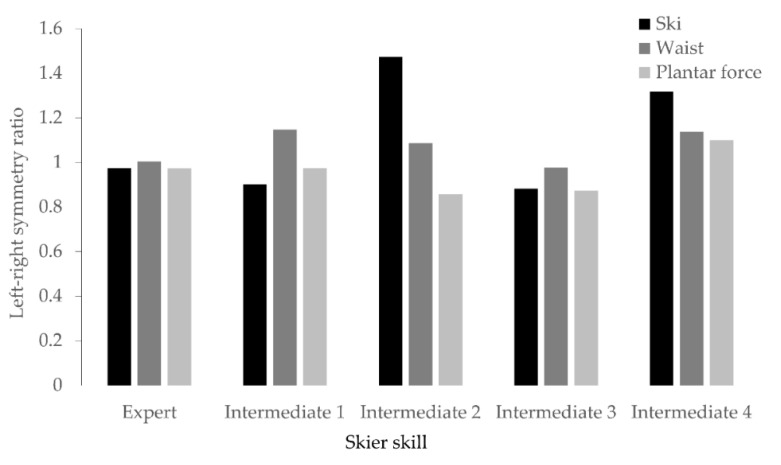
Left–right feature symmetry. The black, dark gray, and light gray bars represent the symmetry ratios of ski motion, waist rotation, and heel load ratio features, respectively. As the value approaches 1, it indicates that the feature has left–right symmetry.

**Table 1 sensors-21-00834-t001:** Results of the post-hoc comparisons for the quantitative ski motion feature. The expert skier was compared to each intermediate skier.

Skier	*t*	*p*	95% CI	Cohen’s *d*
Intermediate skier 1	3.580	0.016 *	4.170 × 10^5^, 9.752 × 10^6^	0.800
Intermediate skier 2	7.797	<0.001 ***	6.408 × 10^6^, 1.574 × 10^7^	1.744
Intermediate skier 3	3.586	0.016 *	4.264 × 10^5^, 9.761 × 10^6^	0.802
Intermediate skier 4	4.761	0.002 **	2.096 × 10^6^, 1.143 × 10^7^	1.065

* *p* < 0.05, ** *p* < 0.01, *** *p* < 0.001.

**Table 2 sensors-21-00834-t002:** Results of the post-hoc comparisons for the quantitative waist rotation feature. The expert skier was compared to each intermediate skier.

Skier	*t*	*p*	95% CI	Cohen’s *d*
Intermediate skier 1	3.398	0.024 *	0.008, 0.505	0.760
Intermediate skier 2	2.255	0.158	−0.078, 0.419	0.504
Intermediate skier 3	3.963	0.010 **	0.051, 0.548	0.886
Intermediate skier 4	7.317	<0.001 ***	0.305, 0.801	1.636

* *p* < 0.05, ** *p* < 0.01, *** *p* < 0.001.

**Table 3 sensors-21-00834-t003:** Results of the post-hoc comparisons for mean heel load ratio. The expert skier was compared to each intermediate skier.

Skier	*t*	*p*	95% CI	Cohen’s *d*
Intermediate skier 1	−6.543	<0.001 ***	−37.771, −12.517	−1.463
Intermediate skier 2	−0.385	1.000	−14.108, 11.147	−0.086
Intermediate skier 3	0.192	1.000	−11.890, 13.365	0.043
Intermediate skier 4	6.514	<0.001 ***	12.402, 37.657	1.456

*** *p* < 0.001.

## Data Availability

We are not able to make our data available to the public because of privacy constraints (in accordance with the consent form used in this experiment).

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
