# Peer review of "Comfortable and Convenient Turning Skill Assessment for Alpine Skiers Using IMU and Plantar Pressure Distribution Sensors"

_sensors, 2021, doi:10.3390/s21030834_

Round 1

Reviewer 1 Report

The authors present a system that uses inertial measurement units (IMU) and plantar pressure distribution sensors to turn skill assessment during alpine skiing. The experiments in real conditions were realized to prove the practical usability of the system. The authors evaluate the system on five alpine skiers, one expert skier and four intermediate skiers.

The paper is well prepared and organized. The text is entirely understandable. The introduction and methodology are clearly described. The results are clearly presented and supplemented with the discussion.

Minor Comments:

  1. The abstract should be slightly rearranged. The quantitative results are completely missing in the abstract. I recommend appending a summary of the essential results directly to the abstract.
  2. line 139 and Figure: Please, indicate the correspondence of x, y and z axes and the body orientation.

Author Response

Dear Reviewer 1,

We really appreciate your helpful and pertinent comments regarding our manuscript, which we have revised accordingly. Our responses to your comments can be found below.

Sincerely,

Seiji Matsumura (corresponding author)

Point 1: The abstract should be slightly rearranged. The quantitative results are completely missing in the abstract. I recommend appending a summary of the essential results directly to the abstract.

Response 1: We added quantitative results to the abstract.

Point 2: line 139 and Figure: Please, indicate the correspondence of x, y and z axes and the body orientation.

Response 2: In section 2.4. Data Analysis, we added a description in the text (line 149 in a revised version) of how edging rotation and the y-axes of the IMUs on the skis correspond to one another; Figure 4 was also modified. Additionally, a sentence was added to refer readers to Figure 2, which shows how the x-, y-, and z-axes correspond to body orientations.

Reviewer 2 Report

This paper presents a wearable system devoted to capture skiers' posture with the aim of improving skills at this sport. The system encompasses two IMUs, one located on the skis and the other on the user's waist, as well as a couple of plantar pressure sensors to capture the skier's motion. Five test skiers, one expert and four intermediate, worn the system. Data revealed the differences in waist rotation and plantar pressure distribution between the expert and the other subjects.  These data could be further used to improve skills of ski enthusiasts and sportsmen.

Overall, this paper addresses an interesting topic of wearable sensing systems applied to sports. The paper is well written; it is easy to read, and to follow. Yet, I recommend a major revision expecting to see the following remarks addressed in the revised version:

A) Content

a.1) the technical description of the monitoring system is indeed sparse. Key details are missing, specially concerning the control unit. It seems that only a memory card is needed to store signals as all processing is conducted off-line.

a.2) the research limits to report the signals collected and to manually compare them between individuals. I wonder if a data fusion is further needed to offer a comprehensive automatic diagnostic and more pertinent feedback to the user. How the three variables, ski, waist, and plantar pressure, combine to impact overall performance.

B) Format

b.1) Equations in Appendix A should be embedded and discussed in the main text.

b.2) Data obtained in "Section 3. Results" should be presented as tables to ease reading: Lines 205-208 and Lines 225-227.

Author Response

Dear Reviewer 2,

We really appreciate your helpful and pertinent comments regarding our manuscript, which we have revised accordingly. We have included responses to your comments below.

Sincerely,

Seiji Matsumura (corresponding author)

A) Content

Point a.1: the technical description of the monitoring system is indeed sparse. Key details are missing, specially concerning the control unit. It seems that only a memory card is needed to store signals as all processing is conducted off-line.

Response a.1: We added technical descriptions of the control and monitoring system (section 2.2. Measurement Sensors), including how we controlled the sensors and extracted the data.

Point a.2: the research limits to report the signals collected and to manually compare them between individuals. I wonder if a data fusion is further needed to offer a comprehensive automatic diagnostic and more pertinent feedback to the user. How the three variables, ski, waist, and plantar pressure, combine to impact overall performance.

Response a.2: We added the limitation (section 4. Discussion) that, an important focus in this study was comfortable measurement; however, from a practical perspective, automatic data collection and comparison should also be considered. Additionally, we discussed how the data from these sensors are combined (related) to measure performance; see section 4. Discussion as well.

B) Format

Point b.1: Equations in Appendix A should be embedded and discussed in the main text.

Response b.1: We embedded all equations in the main text in section 2.4. Data Analysis.

Point a.2: Data obtained in "Section 3. Results" should be presented as tables to ease reading: Lines 205-208 and Lines 225-227.

Response b.2: The results of the post hoc tests for each feature are now presented as tables in section 3. Results.

Round 2

Reviewer 2 Report

The paper has undoubtedly improved from its previous version. I appreciate that my remarks were taken into account. In particular, the technical description of the wearable device is now complete and the acknowledgement of the manual processing of the signals. Authors should consider that the next natural step in this project is data fusion and automatic diagnosis. Tables 1 to 3 definitively ease data comparison.

I have no further content comments or mayor remarks; therefore, I recommend now this paper's acceptance.